# Observational Variables for Considering a Switch from a Normal to a Dysphagia Diet among Older Adults Requiring Long-Term Care: A One-Year Multicenter Longitudinal Study

**DOI:** 10.3390/ijerph19116586

**Published:** 2022-05-28

**Authors:** Maaya Takeda, Yutaka Watanabe, Takae Matsushita, Kenshu Taira, Kazuhito Miura, Yuki Ohara, Masanori Iwasaki, Kayoko Ito, Junko Nakajima, Yasuyuki Iwasa, Masataka Itoda, Yasuhiro Nishi, Junichi Furuya, Yoshihiko Watanabe, George Umemoto, Masako Kishima, Hirohiko Hirano, Yuji Sato, Mitsuyoshi Yoshida, Yutaka Yamazaki

**Affiliations:** 1Gerodontology, Department of Oral Health Science, Faculty of Dental Medicine, Hokkaido University, Sapporo 060-8586, Japan; takedamaaya@den.hokudai.ac.jp (M.T.); tkea-224@den.hokudai.ac.jp (T.M.); kenshu.taira@den.hokudai.ac.jp (K.T.); kmiura@den.hokudai.ac.jp (K.M.); yutaka8@den.hokudai.ac.jp (Y.Y.); 2Tokyo Metropolitan Institute of Gerontology, Tokyo 173-0015, Japan; yohara@tmig.or.jp (Y.O.); iwasaki@tmig.or.jp (M.I.); h-hiro@gd5.so-net.ne.jp (H.H.); 3Oral Rehabilitation, Niigata University Medical and Dental Hospital, Niigata 951-8520, Japan; k-ito@dent.niigata-u.ac.jp; 4Department of Oral Medicine and Hospital Dentistry, Tokyo Dental College, Tokyo 272-8513, Japan; jun.k.nakajima@gmail.com; 5Department of Dentistry, Haradoi Hospital, Fukuoka 813-8588, Japan; y_iwasa@haradoi-hospital.com; 6Department of Oral Rehabilitation, Osaka Dental University Hospital, Osaka 573-1144, Japan; m.itoda89@gmail.com; 7Department of Oral and Maxillofacial Prosthodontics, Kagoshima University Graduate School of Medical and Dental Sciences, Kagoshima 890-8544, Japan; shar@dent.kagoshima-u.ac.jp; 8Department of Geriatric Dentistry, Showa University School of Dentistry, Tokyo 145-8515, Japan; furuya-j@dent.showa-u.ac.jp (J.F.); sato-@dent.showa-u.ac.jp (Y.S.); 9Department of Healthcare Management, Tohoku Fukushi University, Sendai 981-8522, Japan; yoshiw@tfu.ac.jp; 10Swallowing Disorders Center, Fukuoka University Hospital, Fukuoka 814-0180, Japan; george@fukuoka-u.ac.jp; 11Wakakusa-Tatsuma Rehabilitation Hospital, Daito 574-0012, Japan; tdbqd500@yahoo.co.jp; 12Department of Dentistry and Oral-Maxillofacial Surgery, School of Medicine, Fujita Health University, Dengakugakubo, 1-98, Kutsukake-cho, Toyoake 470-1192, Japan; mitsuyoshi.yoshida@fujita-hu.ac.jp

**Keywords:** food form, eating/swallowing functions, dysphagia diet, long-term care facility, tongue movement, perioral muscle function, rinsing

## Abstract

This one-year multicenter longitudinal study aimed to assess whether older adult residents of long-term care facilities should switch from a normal to a dysphagia diet. Using the results of our previous cross-sectional study as baseline, older adults were subdivided into those who maintained a normal diet and those who switched to a dysphagia diet. The explanatory variables were age, sex, body mass index (BMI), Barthel Index, clinical dementia rating (CDR), and 13 simple and 5 objective oral assessments (remaining teeth, functional teeth, oral diadochokinesis, modified water swallowing test, and repetitive saliva swallowing test), which were used in binomial logistic regression analysis. Between-group comparison showed a significantly different BMI, Barthel Index, and CDR. Significant differences were also observed in simple assessments for language, drooling, tongue movement, perioral muscle function, and rinsing and in objective assessments. In multi-level analysis, switching from a normal to a dysphagia diet was significantly associated with simple assessments of tongue movement, perioral muscle function, and rinsing and with the objective assessment of the number of functional teeth. The results suggest that simple assessments can be performed regularly to screen for early signs of discrepancies between food form and eating/swallowing functions, which could lead to the provision of more appropriate food forms.

## 1. Introduction

The prevalence of dysphagia among older adults in Japan is expected to increase, as the country enters a “super-aging” phase [1]. According to a recent Japanese study, the prevalence of dysphagia among healthy older adults living in the community and in nursing homes is 25.1% and 53.8%, respectively [2]. Providing the appropriate food form to patients with dysphagia can help prevent aspiration, asphyxia, and undernutrition as well as maintain quality of life (QOL) [3,4]. However, delayed transition to the appropriate food form due to unnoticed reduced eating/swallowing functions in older adults can increase their risks of preventable complications of dysphagia. 

Video fluorography (VF) and video endoscopy (VE) performed by dysphagia specialists are important tools for evaluating the eating/swallowing functions and decide what food form to provide [5,6]. However, these procedures are difficult to perform frequently at home, in certain medical institutions, and in nursing institutions [7]. 

Caregivers (e.g., nurses, nursing care staff, and family members) may be able to notice a decline in the eating/swallowing functions in older adults who require nursing care for daily activities such as meals, conversations, and oral hygiene. However, if the reduced eating/swallowing functions are mainly due to aging or disuse, the deterioration is often gradual, and the changes are minimal and, thus, can go unnoticed [8]. 

Promptly detecting discrepancies between eating/swallowing functions and food form through examination at a medical institution specializing in dysphagia and subsequently providing the appropriate food form based on the evaluation is deemed highly important in preventing aspiration, asphyxia, and undernutrition and for maintaining QOL [3]. 

Previously, we compared older adults requiring nursing care in Japan who were on a normal diet (ND group) with those on a dysphagia diet (DD group) [9]. We conducted a cross-sectional survey based on the hypothesis that non-specialist caregivers could screen food forms through a simple assessment of the eating/swallowing functions that were readily observable, including whether the patient was choking or engaging in rinsing. Therefore, based on the hypothesis that simple assessments could be used to predict switching from a normal to a dysphagia diet, we conducted a one-year prospective multicenter longitudinal study among long-term care facility (LTCF) residents in Japan. We aimed to identify variables that could be used to predict a switch from a normal to a dysphagia diet. 

## 2. Materials and Methods

### 2.1. Study Design and Participants

This was a one-year prospective multicenter longitudinal study on older adults who were LTCF residents in Japan. We conducted a training session for 30 members of a special committee of the Japanese Society of Gerodontology to explain the study and to standardize the evaluation criteria of the survey. Each member explained the study to the directors and staff of the LTCF they worked with, and eventually 37 LTCF in 17 regions of Japan agreed to participate. The survey was conducted from October 2018 to February 2019. In September 2019, we asked the 37 facilities that had participated in the previous year’s study to take part in another survey. Twenty-five facilities agreed, and 431 residents who had completed the previous year’s study agreed to participate. The study was approved by the ethics committees of the Japanese Society of Gerodontology (2018-1) and the Hokkaido University Faculty of Dental Medicine (2020 No. 4). Written informed consent was obtained from the 455 residents who had participated in the first year’s survey [10]. The survey was conducted in the same manner as the baseline survey (Figure 1).

### 2.2. Survey Items

Before the survey, the study members conducted a training session for all the nurses and registered dietitians at their respective facilities on how to evaluate the survey items to standardize the assessment criteria. Thereafter, the survey forms were distributed to the nurses and registered dietitians at each facility to assess the resident participants.

#### 2.2.1. Questionnaire Survey

##### Basic Information

The registered dietitians obtained the residents’ age, sex, and body mass index (BMI) from the nursing care records. The BMI was categorized as follows: 0, ≥18.5 kg/m^2^; 1, <18.5 kg/m^2^ [11]. 

##### Life and Cognitive Function Assessment

The nurses conducted a basic assessment of activities of daily living using the Barthel Index (BI) [12]. Cognitive function was evaluated using the Clinical Dementia Rating (CDR) based on the method of Morris et al. [13]. The CDR determines the severity of dementia on a five-point scale as follows: 0, healthy; 0.5, suspected dementia; 1, mild dementia; 2, moderate dementia; 3, severe dementia. One Japanese psychiatrist specializing in dementia and certified as a dementia specialist in Japan checked all assessments and determined the final CDR score.

#### 2.2.2. Oral Status

The nurses assessed the status of the oral cavity based on a food form survey conducted in advance by the LTCF staff, which was collated by the investigators. A manual was used to explain the assessments to the nurses. The nurses were accompanied by an investigator to evaluate 4–5 residents to ensure that the assessment criteria were standardized. 

Language was assessed in three grades: 0, able to speak; 1, able to speak but with poor articulation; and 2, unable to speak. An assessment of 0 was classified as “good”, while that of 1 or 2 was classified as “poor”. Drooling was assessed in three grades: 0, never; 1, sometimes; and 2, always. An assessment of 0 was classified as “good”, while that of 1 or 2 was classified as “poor”. Halitosis was assessed in three grades: 0, none; 1, slight, and 2, severe. An assessment of 0 was classified as “good”, while that of 1 or 2 was classified as “poor”. Tongue movement was assessed in three grades: 0, nearly complete movement; 1, small range of motion; and 2, no movement. An assessment of 0 was classified as “good”, while that of 1 or 2 was classified as “poor”. Perioral muscle function was assessed in three grades: 0, movement; 1, slight difficulty; and 2, no movement. An assessment of 0 was classified as “good”, while that of 1 or 2 was classified as “poor”. Left–right asymmetric movement of the mouth angle was assessed in two grades: 0, no; and 1, yes. Rinsing was assessed in three grades: 0, capable; 1, capable but imperfect; and 2, incapable. An assessment of 0 was classified as “good”, while that of 1 or 2 was classified as “poor”.

Masticatory movement was assessed in three grades: 0, has movement; 1, moves when spoken to; and 2, almost no movement. An assessment of 0 was classified as “good”, while that of 1 or 2 was classified as “poor”. Swallowing was assessed in two grades: 0, capable, and 1, delayed but capable. Coughing was assessed in two grades: 0, no coughing, and 1, coughing. Changes in voice quality after swallowing was assessed in two grades: 0, no, and 1, yes. Respiratory observation after swallowing was assessed in two grades: 0, no abnormalities, and 1, becomes shallow and fast. Oral residue was assessed in three grades: 0, none; 1, a small amount; and 2, present. An assessment of 0 was classified as “good”, while that of 1 or 2 was classified as “poor”. The oral condition was assessed by observing it after ordinary meals.

Food form was classified using the Japanese Society of Dysphagia Rehabilitation’s 2013 dysphagia diet classification codes [14,15]. The dietitian in charge identified diets classified as dysphagia diets. The nature of the diet and its contents, whether solid or liquid, depended on each subject. This is because the diet being consumed by the residents at the LTCFs was not prepared for the purpose of this study; it was the diet that participants usually ate. Diets containing food cooked or modified to be soft and chewable with weak force and food that did not require special cooking or modification were classified as normal diets.

#### 2.2.3. Measurements

The surveys were conducted by 30 dentists and dental hygienists who were trained in advance and used standardized evaluation criteria. 

##### Oral Status Assessment

The number of functional teeth was the sum of the number of remaining teeth and prosthetic teeth (e.g., implants, pontics, dentures). 

##### Objective Assessment of the Oral Function

Oral diadochokinesis (ODK)

This test is a comprehensive measurement of the motor dexterity of the tongue and lips. The residents repeated the sounds /pa//ta//ka/ for 5 s to measure the number of times each syllable was spoken per second using an automatic measuring instrument (Kenkou-kun Handy, Takei Scientific Instruments Co., Niigata, Japan). The cutoff values for /ta/ used in this analysis were 5.2 and 5.4 for men and women, respectively. Therefore, men were assessed as 0: <5.2, 1: ≥5.2, and women as 0: <5.4, 1: ≥5.4 [16]. 

2.Modified water swallowing test (MWST)

The cervical auscultation method [17] was used with the MWST [18] to evaluate the swallowing function. Following the normal method, 3 mL of cold water was poured into the floor of the oral cavity using a 5 mL syringe, and the resident was instructed to swallow. A stethoscope was then used to evaluate changes in swallowing and respiratory sounds before and after swallowing. An abnormality was considered present (0, abnormal; 1, normal) if there were wet or foamy sounds during pharyngeal swallow or if wheezing or a cough reflex was observed [17]. 

3.Repetitive saliva swallowing test (RSST)

The RSST [19] was used to evaluate the swallowing function. The investigator instructed the residents to perform as many empty swallows as possible in 30 s in a sitting position. The investigator placed the index and middle fingers on the resident’s hyoid bone and laryngeal prominence and counted the number of times the hyoid bone moved above the investigator’s fingers during the swallowing reflex. The total number of times was recorded. The results were classified as 0, total <3 and 1, total ≥3. 

### 2.3. Statistical Analysis

The residents were first classified into a group with poor nutritional intake status (<75% of the mean) and a group with good nutritional intake status (≥75% of the mean). The reference values denote that the average daily dietary intake is <75% or ≥75% of the diet provided by a registered dietitian based on a previous study on dysphagia [9]. Residents with poor nutritional status were excluded because they might be provided with food forms that were not suitable for the assessment of eating and swallowing function. The good group was then subdivided into a group that was on a dysphagia diet at baseline (DD group) and a group that was on a normal diet at baseline (ND group). In addition, the ND group was subdivided into a group that was on an ND one year after baseline (ND maintenance group) and a group that was on a DD (DD switched group). The baseline survey items were compared between the two groups. The comparisons between the groups in terms of sex, simple assessments of oral status, and CDR were analyzed using the chi-square test. Continuous variables were tested for normality; subsequently, an unpaired t-test was used for age and BMI, and the Mann–Whitney U test was used for BI, remaining teeth, functional teeth, ODK, MWST, and RSST. 

As data from multiple institutions were analyzed, a random effect from the institution was confirmed using multi-level analysis. To examine the factors associated with switching from a normal diet to a dysphagia diet, a multi-level analysis was performed on the 251 residents in the ND group, with staying on an ND one year after baseline or switching to a DD as the dependent variable to calculate the odds ratio (OR) and 95% confidence interval (CI). 

The explanatory variables were age and sex, and the covariates were BMI [20], BI [21], and CDR [4], which have been reported to be associated with food form. In addition, remaining teeth, functional teeth, ODK, RSST, and MWST were used as explanatory variables for the objective oral assessments. The explanatory variables of the simple oral assessments included language, drooling, halitosis, masticatory movement, tongue movement, perioral muscle function, left–right asymmetric movement of the mouth angle, swallowing, coughing, changes in voice quality after swallowing, respiratory observation after swallowing, rinsing, and presence of oral residues. All statistical analyses were performed using SPSS Software version 26 (IBM Corp, Armonk, NY, USA), and the significance level was set at 5% (*p* < 0.05). 

## 3. Results

A total of 431 LTCF residents (84 men, 347 women, mean age 87.4 ± 7.9 years) participated in the baseline survey and the survey one year later. Of the 400 residents in the good nutritional intake group, 149 (37.3%) were on a dysphagia diet at baseline in 2018, and 251 (62.7%) were on a normal diet. The ND group had a significantly lower age and CDR, significantly higher BMI and BI, and more functional teeth than the DD group. In addition, this group had significantly higher proportions classified as “good” for all the simple and objective assessment items (Table 1). 

Of the 251 residents on a normal diet in 2018, 47 (18.7%) had switched to a dysphagia diet in 2019, and 204 (81.3%) remained on a normal diet. The ND maintenance group had a significantly lower CDR and significantly higher BMI and BI than the DD-switched group. Moreover, the ND maintenance group had significantly higher proportions classified as “good” in the following simple assessments: language, drooling, tongue movement, perioral muscle function, and rinsing. Moreover, the ND maintenance group had significantly higher proportions classified as “good” in the following objective assessments: number of functional teeth, ODK, RSST, and MWST (Table 2). 

A multi-level analysis supported the assumption that the characteristics of the participants would differ between institutions. Therefore, the group that was on a normal diet in 2018 was classified into two groups—those who maintained a normal diet (ND maintenance group), and those who switched to a dysphagia diet (DD-switched group) in 2019. These were used as the dependent variables in a multi-level analysis. 

First, analysis of each of the simple assessments individually showed that the following were significantly associated with the distinction between a normal and a dysphagia diet: language (OR: 1.02, 95% CI: 0.02–0.25), tongue movement (OR: 1.14, 95% CI: 0.13–0.38), perioral muscle function (OR: 1.13, 95% CI: 0.12–0.42), swallowing (OR: 1.01, 95% CI: 0.01–0.40), coughing (OR: 1.00, 95% CI: 0.00–0.24), and rinsing (OR: 1.11, 95% CI: 0.10–0.32). In contrast, significant differences were only observed for the objective evaluation of the number of functional teeth (OR: 1.00, 95% CI: 0.00–0.02) (Table 3: Model 1). 

Binomial logistic analyses of individual adjustment variables (age, sex, BMI, BI, CDR), with the two categories of maintaining a normal diet and switching to a dysphagia diet as the dependent variable, found significant differences in BMI, BI, and CDR (Table 4). 

Thus, when the BMI, BI, and CDR were used to analyze the simple assessments, significant differences were observed in tongue movement (OR: 1.06, 95% CI: 0.06–0.31), perioral muscle function (OR: 1.06, 95% CI: 0.06–0.36), and rinsing (OR: 1.01, 95% CI: 0.01–0.25). In contrast, significant differences were only observed for the objective evaluation of the number of functional teeth (OR: 1.00, 95% CI: 0.00–0.01) (Table 3: Model 2). 

The age, BMI, BI, and CDR were then used as the adjustment variables to analyze each of the simple assessments; significant differences were found in tongue movement (OR: 1.06, 95% CI: 0.06–0.31), perioral muscle function (OR: 1.05, 95% CI: 0.05–0.36), and rinsing (OR: 1.01, 95% CI: 0.01–0.25). Moreover, a significant difference was observed in the objective evaluation of the number of functional teeth (OR: 1.00, 95% CI: 0.00–0.01) (Table 3: Model 3). 

Models 1, 2, and 3 showed significant differences for the simple assessments of tongue movement, perioral muscle function, and rinsing, but only for the objective evaluation of the number of functional teeth.

## 4. Discussion

The objective of this study was to identify observational items that can predict switching from a normal to a dysphagia diet. The results showed that simple assessments that caregivers can observe in daily activities—tongue motions, perioral muscle functions, and rinsing—as well as objective evaluations that can be performed by a dentist or other specialist—number of functional teeth—were associated with switching from a normal to a dysphagia diet. The simple assessments were items that could be observed and evaluated by caregivers who are familiar with older adults requiring nursing care for everyday activities such as dietary assistance and oral care. If changes in these assessments are in fact signs that can predict switching from a normal to a dysphagia diet, this method would be easy to disseminate in nursing care settings and could be used to help decide whether to refer older adults who require nursing care to a medical institution specializing in dysphagia, subsequently preventing undernutrition, aspiration, and asphyxiation. 

In our previous cross-sectional study that considered on older adults on a normal and a dysphagia diet as the dependent variable, the simple assessments of coughing and rinsing and the objective evaluations of the number of teeth, number of functional teeth, and RSST were associated with the distinction between a normal and a dysphagia diet. The present study was a longitudinal survey, and we consider the findings beneficial, as the dependent variable was maintaining a normal diet or switching from a normal to a dysphagia diet for each resident. 

Coughing, which was associated with food form in the cross-sectional study, is a finding of the pharyngeal stage in the five-phase model of eating and swallowing movements described by Leopold et al., and rinsing is a finding of the preparation stage [22]. Rinsing was significant, possibly because it could be assessed in oral care situations, such as gargling, that are unrelated to food form. 

Both tongue movement and perioral muscle function that exhibited significant differences in the present study are preparation stage findings. As these are muscle movements involved in mastication and are related to the food form, the result seems reasonable. 

One factor that may have contributed to coughing not showing a significant association in the present longitudinal study is that in the group that switched to a dysphagia diet in 2019, the swallowing function may have declined, and the cough reflex may have been impaired. Therefore, even if the swallowing function declines, if the residents do not cough, it is impossible to make an assessment based on whether the person coughed. 

The objective evaluations ODK and RSST, which did not exhibit significant associations in the cross-sectional study, also did not exhibit significant associations in the present longitudinal study. Further, while the MWST exhibited a significant association in the cross-sectional study, it was only significant in Model 1 in the present study. As discussed in the cross-sectional study, for ODK and RSST, the residents should understand and be willing to perform the tests. Comparing the CDR between the residents of the previous study and those of the present study, we observed that in the previous study, 9.3% of the residents had a CDR of 0 or 0.5, and 19.9% had a CDR of 1, while in the present study, 15.9% had a CDR of 0 or 0.5, and 27.9% had a CDR of 1, indicating that even if the group had a high proportion of participants with relatively good cognitive function, fewer participants were able to perform the test; even among those who could perform the test, only few understood its purpose or had the motivation to perform it. When water is placed in the oral cavity, the MWST can assess instinctive swallowing movements, regardless of the patient’s understanding of or willingness to perform the test. While the MWST can reportedly detect dysphagia with high sensitivity and specificity using small amounts of water [18], a significant difference was not observed in the present longitudinal study. 

The number of functional teeth was significantly different in both the present longitudinal study and the cross-sectional study, which provides supporting evidence for the previous study. The simple assessments performed in the present study were only explained in advance to the nurses by an investigator using a manual, then the criteria were standardized by performing the evaluations together with an investigator on 4–5 residents; nonetheless, valid results were still obtained. This indicates that a valid result can be obtained without special training, and disseminating this approach is both easy and beneficial. 

VE [23,24] and VF [25], the gold standards for conducting a thorough examination of the eating/swallowing functions, could not be performed in the present study [23,24]. However, because these tests are performed in unusual settings, they do not necessarily assess everyday eating/swallowing functions. Further, it is challenging to perform these tests frequently at an LTCF. 

Therefore, if a discrepancy between a patient’s everyday eating/swallowing functions and food form can be detected by the simple observational items that exhibited significant differences in the present study, this could serve as an effective screening tool for referral to a specialist or specialized medical institution. 

We hypothesized that if the nutritional intake status was good, food forms suitable to residents’ eating/swallowing functions would be provided. This is because if food forms that go beyond their eating/swallowing functions are provided, the residents are unable to eat them, which would result in the deterioration of their nutritional intake status. In fact, the group with a good nutritional intake status had a significantly higher BMI and more functional teeth than the group with a poor status. Moreover, the good group had significantly higher proportions with good results in several of the simple assessments, including tongue and perioral muscle function. Based on these findings, we believe that residents with a good nutritional intake status are unlikely to be given food forms that go beyond their functional abilities. 

When comparing the groups that were on a normal and a dysphagia diet in 2018, significant differences were observed in all items except sex. This result also suggests that food forms suitable to the residents’ eating/swallowing functions were being provided. 

It is known that when dementia progresses, appetite and food intake decrease. It has also been found that this is preceded by changes in eating behavior due to the progression of dementia, reduced dietary independence, and dysphagia [26]. These changes have been reported to reduce food intake, leading to undernutrition, dehydration, a poorer general condition, and decreased immune and cognitive functions, resulting in an increased risk of aspiration pneumonia and death [26,27]. Another study found that when the eating/swallowing functions decline, in addition to the risk of undernutrition, the risk of suffocation and aspiration also increases [28]. Thus, it is possible to modify the food form so the person can continue eating safely and enjoy meals [29]. 

The appearance of undernutrition due to decreased eating/swallowing functions in older adults who require nursing care has been reported to substantially affect the severity of their nursing care needs and survival prognosis [30].

In an LTCF, the prevention of incidents such as asphyxiation and aspiration is prioritized over trying to maintain food forms, and many cases of switching to a dysphagia diet have been reported [10,31]. However, Endo et al. reported that switching from a normal to a dysphagia diet is associated with weight loss in LTCF residents [10], which suggests that the food form should not be changed hastily without any evaluation. The findings that predict switching from a normal to a dysphagia diet as revealed by the present study could be an effective tool to maintain dietary safety and appetite in older adults with cognitive decline who require a high level of nursing care and may help prevent undernutrition, suffocation, aspiration, and other conditions in LTCF that do not have in-house specialists. 

This study has some limitations. First, it should be noted that the institutions in this study are affiliated with members of the Japanese Society of Gerodontology, and thus there may be bias in the sampling. Next, the presence/absence and content of dental therapies that could have caused the onset of new diseases, exacerbated comorbidities, or affected oral intake during the study period were not considered. However, as these are highly individualized and infrequent, we believe they had minimal impact on the results. In addition, it should be noted that the results of this study were obtained from people with a good nutritional intake. Lastly, we did not conduct swallowing endoscopy or swallowing angiography, which are the gold standards for assessing the swallowing function. While these examinations are difficult to perform in a multi-center study with many participants, we believe a more detailed study that includes these examinations should be conducted in the future.

## 5. Conclusions

Our results suggest that items such as tongue movement and perioral muscle function can predict switching from a normal to a dysphagia diet. If these simple assessments could be performed regularly by nursing care workers to screen for early signs of discrepancies between food form and eating/swallowing functions, it could help prevent undernutrition, pneumonia, asphyxia, and aspiration in older adults with dysphagia who require nursing care. Going forward, we intend to take the simple assessments examined in this study to nursing care settings to verify their effects.

## Figures and Tables

**Figure 1 ijerph-19-06586-f001:**
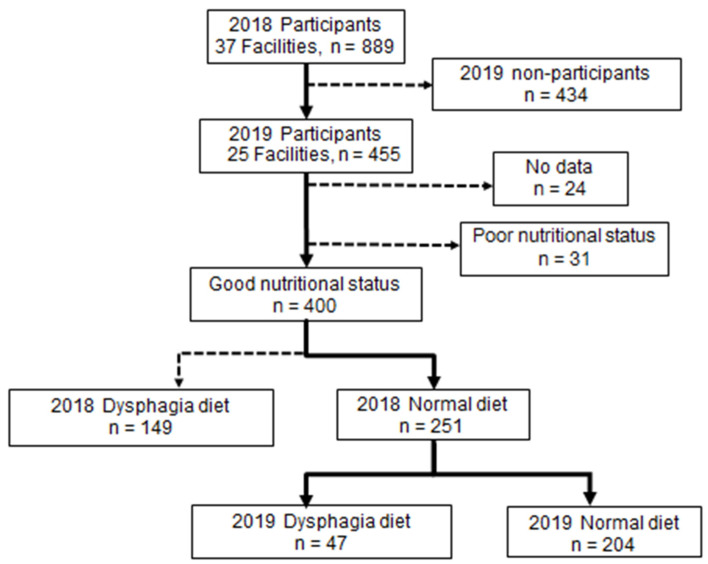
Flow chart of study participation.

**Table 1 ijerph-19-06586-t001:** Comparison of characteristics of the study participants in the 2018 Normal and Dysphagic Diet groups.

Variable	2018 Dysphagia Diet (n = 149)				2018 Normal Diet (n = 251)				Good Nutritional Status (n = 400)				*p*-Value
	Mean ± SD			Median, [Q1, Q3]	Mean ± SD			Median, [Q1, Q3]	Mean ± SD			Median, [Q1, Q3]	
	n (%)				n (%)				n (%)				
Age	87.5	±	7.6	89.0 [83.0, 93.0]	85.9	±	7.8	86.5 [81.0, 92.0]	86.5	±	7.7	87.0 [82.0, 92.0]	0.042
Sex (female), n (%)	120		(80.5)		201		(80.1)		321		(80.3)		0.912
Body mass index	19.4	±	2.7	19.4 [17.4, 21.3]	21.9	±	3.7	21.6 [19.5, 24.1]	21.0	±	3.6	20.8 [18.5, 22.9]	<0.001
Barthel Index (Total points)	17.7	±	19.4	10.0 [0.0, 30.0]	44.0	±	24.8	45.0 [25.0, 60.0]	34.2	±	26.2	35.0 [10.0, 55.0]	<0.001
Clinical dementia rating (Total points)													
0, 0.5	4		(2.7)		40		(15.9)		44		(11.0)		<0.001
1	14		(9.4)		70		(27.9)		84		(21.0)		
2	36		(24.2)		84		(33.5)		120		(30.0)		
3	90		(60.2)		56		(22.3)		146		(36.5)		
Simple evaluations (Oral conditions)													
Language (possible)	80		(53.7)		203		(80.9)		283		(70.8)		<0.001
Drooling (none)	87		(58.4)		222		(88.4)		309		(77.3)		<0.001
Halitosis (none)	88		(59.1)		178		(70.9)		266		(66.5)		0.015
Masticatory movement (move)	116		(77.9)		241		(96.0)		357		(89.3)		<0.001
Tongue movement (move)	76		(51.0)		207		(82.5)		283		(70.8)		<0.001
Perioral muscle function (move)	102		(68.5)		221		(88.0)		323		(80.8)		<0.001
Left–right asymmetric movement of the mouth angle (not)	116		(77.9)		229		(91.2)		345		(86.3)		<0.001
Swallowing (possible)	91		(61.1)		235		(93.6)		326		(81.5)		<0.001
Coughing (not)	57		(38.3)		201		(80.1)		258		(64.5)		<0.001
Changes in voice quality after swallowing (not)	115		(77.2)		237		(94.4)		352		(88.0)		<0.001
Respiratory observation after swallowing (No abnormality)	140		(94.0)		247		(98.4)		387		(96.8)		0.015
Rinsing (possible)	48		(32.2)		196		(78.1)		244		(61.0)		<0.001
Oral residue (none)	53		(35.6)		137		(54.6)		190		(47.5)		<0.001
Objective evaluation of oral function													
Remaining Teeth	6.0	±	7.7	3.0 [0.0, 10.8]	10.1	±	9.5	7.0 [0.0, 19.0]	8.6	±	9.1	5.0 [0.0, 17.0]	<0.001
Functional teeth	16.4	±	11.5	21.5 [3.3, 28.0]	23.3	±	7.5	28.0 [21.0, 28.0]	20.7	±	9.8	27.0 [15.5, 28.0]	<0.002
ODK (ta)	2.9	±	2.9	3.0 [0.95, 3.8]	3.9	±	2.0	4.0 [2.4, 5.0]	3.6	±	2.3	3.6 [2.2, 5.0]	<0.003
RSST	1.9	±	1.2	2.0 [1.0, 3.0]	2.7	±	1.4	3.0 [2.0, 4.0]	2.6	±	1.4	3.0 [2.0, 3.0]	<0.004
MWST	2.0	±	2.1	0.0 [0.0, 4.0]	3.8	±	1.7	4.0 [4.0, 5.0]	3.1	±	2.1	4.0 [0.0, 5.0]	<0.005

SD = standard deviation, Q = quartile, ODK = oral diadochokinesis, RSST = Repetitive saliva swallowing test, MWST = modified water swallowing test.

**Table 2 ijerph-19-06586-t002:** Comparison of characteristics of the study participants in the 2019 Normal and Dysphagic Diet groups.

Variable	2019 Dysphagia Diet (n = 47)				2019 Normal Diet Maintained (n = 204)				*p*-Value
	Mean ± SD			Median, [Q1, Q3]	Mean ± SD			Median, [Q1, Q3]	
	n (%)				n (%)				
Age	86.3	±	7.6	86.0 [81.0, 93.0]	85.8	±	7.8	87.0 [81.0, 92.0]	0.853
Sex (female), n (%)	35		(74.5)		166		(81.4)		0.285
Body mass index	20.8	±	3.7	20.7 [18.3, 23.0]	22.1	±	3.6	21.8 [19.7, 24.3]	0.029
Barthel Index (Total points)	35.5	±	20.0	35.0 [25.0, 50.0]	46.0	±	25.4	45.0 [25.0, 65.0]	0.009
Clinical Dementia Rating (Total points)									
0, 0.5	1		(2.1)		39		(19.1)		0.036
1	14		(29.8)		56		(27.5)		
2	20		(42.6)		64		(31.4)		
3	11		(23.4)		45		(22.1)		
Simple evaluations (oral conditions)									
Language (possible)	33		(70.2)		170		(83.3)		0.039
Drooling (none)	39		(83.0)		183		(89.7)		0.16
Halitosis (none)	30		(62.8)		148		(72.5)		0.235
Masticatory movement (move)	44		(93.6)		197		(96.6)		0.351
Tongue movement (move)	30		(63.8)		177		(86.8)		<0.001
Perioral muscle function (move)	35		(74.5)		186		(91.2)		0.003
Left–right asymmetric movement of the mouth angle (not)	42		(89.4)		187		(91.7)		0.936
Swallowing (possible)	41		(87.2)		194		(95.1)		0.047
Coughing (not)	34		(72.3)		167		(81.9)		0.141
Changes in voice quality after swallowing (not)	42		(89.4)		195		(95.6)		0.094
Respiratory observation after swallowing (No abnormality)	45		(95.7)		202		(99.0)		0.106
Rinsing (possible)	29		(61.7)		167		(81.9)		0.003
Oral residue (none)	23		(48.9)		114		(55.9)		0.37
Objective evaluation of oral function									
Remaining teeth	9.0	±	9.2	5.0 [0.0, 18.3]	10.3	±	9.5	8.0 [0.0, 19.0]	0.475
Functional teeth	20.9	±	9.5	25.0 [16.3, 28.0]	23.8	±	6.9	28.0 [22.0, 28.0]	0.08
ODK (ta)	4.4	±	1.4	4.6 [3.4, 5.4]	3.8	±	2.1	3.8 [2.4, 5.0]	0.03
RSST	2.0	±	1.2	2.0 [1.0, 3.0]	3.0	±	1.4	3.0 [2.0, 4.0]	0.003
MWST	3.3	±	1.9	4.0 [2.3, 5.0]	4.0	±	1.6	4.0 [4.0, 5.0]	0.011

SD = standard deviation, Q = quartile, ODK = oral diadochokinesis, RSST = Repetitive saliva swallowing test, MWST = modified water swallowing test.

**Table 3 ijerph-19-06586-t003:** Results of the multi-level analysis of simple evaluation (oral conditions) and objective evaluation of the oral function.

	Model 1					Model 2					Model 3				
**Oral Status**	**OR**		**95% CI**			**OR**		**95% CI**			**OR**		**95% CI**		
Language (1: good, 2: bad)	1.02	*	0.02	–	0.25	0.93		−0.08	–	0.17	0.93		−0.07	–	0.17
Drooling (1: no, 2: yes)	0.94		−0.06	–	0.24	0.89		−0.12	–	0.18	0.89		−0.12	–	0.18
Halitosis (1: no, 2: yes)	0.94		−0.06	–	0.15	0.88		−0.13	–	0.08	0.88		−0.13	–	0.08
Masticatory movement (1: good, 2: bad)	0.97		−0.03	–	0.45	0.93		−0.07	–	0.39	0.93		−0.07	–	0.40
Tongue movement (1: good, 2: bad)	1.14	**	0.13	–	0.38	1.06	*	0.06	–	0.31	1.06	*	0.06	–	0.31
Perioral muscle function (1: good, 2: bad)	1.13	**	0.12	–	0.42	1.06	*	0.06	–	0.36	1.05	*	0.05	–	0.36
Left–right asymmetric movement of the mouth angle (1: good, 2: bad)	0.80		−0.22	–	0.11	0.77		−0.26	–	0.05	0.77		−0.26	–	0.05
Swallowing (1: good, 2: bad)	1.01	*	0.01	–	0.40	0.91		−0.09	–	0.29	0.91		−0.09	–	0.29
Coughing (1: no, 2: yes)	1.00	*	0.00	–	0.24	0.94		−0.07	–	0.17	0.94		−0.06	–	0.18
Changes in voice quality after swallowing (1: no abnormality, 2: abnormality)	0.92		−0.08	–	0.33	0.84		−0.17	–	0.23	0.84		−0.17	–	0.24
Respiratory observation after swallowing (1: good, 2: bad)	0.88		−0.13	–	0.60	0.84		−0.17	–	0.52	0.85		−0.16	–	0.54
Rinsing (1: possible, 2: impossible)	1.11	**	0.10	–	0.32	1.01	*	0.01	–	0.25	1.01	*	0.01	–	0.25
Oral residue (1: no, 2: yes)	0.99		−0.01	–	0.19	0.93		−0.08	–	0.13	0.93		−0.07	–	0.14
**Objective evaluation of oral function**	**OR**		**95% CI**			**OR**		**95% CI**			**OR**		**95% CI**		
Remaining teeth	1.00		0.00	–	0.01	1.00		0.00	–	0.01	1.00		0.00	–	0.01
Functional teeth	1.00	*	0.00	–	0.02	1.00	*	0.00	–	0.01	1.00	*	0.00	–	0.01
ODK (ta) male	0.73		−0.32	–	0.26	0.66		−0.42	–	0.23	0.66		−0.42	–	0.24
ODK (ta) female	0.78		−0.24		0.04	0.78		−0.25		0.04	0.77		−0.26		0.04
RSST	0.70		−0.36	–	0.18	0.84		−0.18	–	0.03	0.93		−0.08	–	0.15
MWST	0.93		−0.07	–	0.22	0.94		−0.06	–	0.16	0.94		−0.06	–	0.16

CI = confidence interval, OR = odds ratio, ODK = oral diadochokinesis, RSST = Repetitive saliva swallowing test, MWST = modified water swallowing test; * *p* < 0.05, ** *p* < 0.01.

**Table 4 ijerph-19-06586-t004:** Adjustment variables.

Variable	OR		95% CI		
Age	1.00		−0.01	–	0.01
Sex (1: male, 2: female)	0.92		−0.08	–	0.15
Body mass index	0.97	*	−0.03	–	0.00
Barthel Index	0.99	**	−0.01	–	0.00
Clinical Dementia Rating					
0, 0.5	Reference				
1	1.09	*	0.08	–	0.36
2	1.16	**	0.15	–	0.43
3	1.14	**	0.13	–	0.45

CI = confidence interval, OR = odds ratio; * *p* < 0.05; ** *p* < 0.01.

## Data Availability

The data presented in this study are available on request from the corresponding author. The data are not publicly available due to ethico-legal restrictions imposed by the Ethics Committee at the Japanese Society of Gerodontology.

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
