# Peer review of "Observational Variables for Considering a Switch from a Normal to a Dysphagia Diet among Older Adults Requiring Long-Term Care: A One-Year Multicenter Longitudinal Study"

_ijerph, 2022, doi:10.3390/ijerph19116586_

Round 1
Reviewer 1 Report
This paper contains valuable content for care workers working in long-term care facilities and their researchers. However, additional information is required for the parts that are lacking in explanation.
2. Materials and Methods
2.1 Study Design and Participants
Information about institutional review board approval and informed consent for this study should be provided.
(Line 91-93) If you have the published article, please add the reference.
2.2.2 Oral status
What was the diet the subjects were eating when their oral condition was assessed? Was it a liquid? Or was it a solid diet?
Was the meal prepared for survey? Or was it the meal they always eat?
2.2.3 Measurements
(Line 166-168) Please add the reference used to determine the ODK cutoff value.
2.3. Statistical analysis
(Line 186-187) What does ‘<75% of the mean’ mean? Is it the amount of energy consumed? Also, please provide the scientific basis for setting this reference value. In addition, you should state why you excluded subjects who were judged to have poor nutritional status.
(Line 204-212) The sentence with the explanatory variable as the subject continues three times. Please organize it in an easy-to-understand manner.
3. Results
Table 1.
Both ‘n = 149’ and ‘n = 251’ are written in italics. If it doesn't make sense, you should use the same format as any other table.
Table 2.
Does the information in the Sex column represent the number of female (%)? Please modify it like table1.
Table 3.
About ‘Simple evaluations (oral conditions)’, it is described as ‘oral status’ in the text. The term should be unified.
About ‘Present teeth’, it is described as ‘remaining teeth’ in the text. The term should be unified.
ODK (ta) is described separately for male and female, but in the text it is described as ODK (OR: 1.273, …). Did you test males and females separately when doing statistical analysis? Which is correct?
Discussion
(Line 360-363) Please add the reference.
Author Response
添付ファイルをご覧ください。

Reviewer 2 Report
Abstract: consider briefly describing what the "objective assessments" are instead of just saying they're objective, or just list them as ODK, MWST and RSST.
Intro: succinct and clear. Some sentences a little too long, consider editing these to be shorter for ease of reading.
Method: can you also add inclusion/exclusion criteria? Based on figure 1, it seems like residents who had poor nutritional status were excluded after the second survey. Why and how was this determined?
Figure 1: please add number of facilities to the second box (2019 participants) just like you did with the first box (2018 participants)
page 3, line 114: please add reference for Morris et al in text and in the reference section at the end.
2.2.2 Oral status: are these ratings based on overall impression or during specific time frames (e.g. during meal times)? It seems like some are overall impressions and some are specific to swallowing. Consider separating this into 2 paragraphs, or perhaps a table for each measure and what 0, 1, and 3 mean for that measure. The table might make everything easier to read and refer back to.
Page 4, line 126: suggest to change "vocalize" to "speak". Vocalize could be any sound that's not language specific, whereas speak/speech denotes that language is used.
Page 151-153: can you rephrase this to make it more clear that soft and chewable foods for weak force were NOT considered normal? Right now the sentence is a little ambiguous.
ODK: were /pa//ta//ka/ measured once or multiple times to get an average for each sound?
line 350-352: this justification goes in some way to explain why those with poor nutritional status were excluded, but I believe you may be missing part of the picture. We don't know why there was poor nutritional status, and the reasons could have been just as relevant in contributing to diet changes, e.g. primary disease diagnosis/progression. Clinically, carers for LTCF residents will also look out for medical reasons for diet changes or the potential need for diet changes. You have acknowledged this in the limitations, but this explanation does not feel entirely justified or clear.
line 381 and 384: you use "oral intake status" here but "nutritional intake" in the rest of the text, please stick to one.
Author Response
添付ファイルをご覧ください。
